# Imagining a Maya Archaeology That Is Anthropological and Attuned to Indigenous Cultural Heritage

Patricia McAnany

Department of Anthropology; University of North Carolina-Chapel Hill; 301 Alumni Bldg.,
Chapel Hill, NC 27516, USA; mcanany@email.unc.edu

**Abstract:** Taking an aspirational approach, this article imagines what Maya Archaeology would be like if it were truly anthropological and attuned to Indigenous heritage issues. In order to imagine such a future, the past of archaeology and anthropology is critically examined, including the emphasis on processual theory within archaeology and the Indigenous critique of socio-cultural anthropology. Archaeological field work comes under scrutiny, particularly the emphasis on the product of field research over the collaborative process of engaging local and descendant communities. Particular significance is given to the role of settler colonialism in maintaining unequal access to and authority over landscapes filled with remains of the past. Interrogation of the distinction between archaeology and heritage results in the recommendation that the two approaches to the past be recognized as distinct and in tension with each other. Past heritage programs imagined and implemented in the Maya region by the author and colleagues are examined reflexively.

**Keywords:** cultural heritage; Maya archaeology; indigenous critique of anthropology; settler colonialism

## 1. Introduction

> Imagine no possessions
> I wonder if you can
> No need for greed or hunger
> A brotherhood of man
> Imagine all the people sharin' all the world. . . .
>
> You may say I'm a dreamer
> But I'm not the only one
> I hope someday you'll join us
> And the world will live as one

John Lennon & Yoko Ono [1]

Here, I take the words of John Lennon and Yoko Ono to heart and imagine a different kind of Maya Archaeology—one that is couched within anthropological and heritage perspectives. In order to get there, I first turn to processual archaeology of the previous century, which is examined for its afterlife and entanglement with the crisis of representation within Socio-cultural Anthropology. The corresponding assertion of rights to self-representation among Indigenous peoples is discussed in terms of its impact on Maya Archaeology. The lenses of archaeology and heritage are argued to be separate but interlinked spaces of practice that exist in tension within each other. I propose that archaeological practice focuses on process, not in terms of theorizing change but as a methodological shift away from a research

practice that primarily is product-focused. As discussed below, this change already infuses research in Australia and Oceania. Borrowing a term from Indigenous scholar Eve Tuck [2] (p. 644) who refers to a "methodology of repatriation," I emphasize an archaeological practice that is attuned to Indigenous and local voices expressed through the idiom of cultural heritage. Reflecting on heritage programs launched by the *Maya Area Cultural Heritage Initiative* and *InHerit: Indigenous Heritage Passed to Present* (www.in-herit.org), I discuss the challenges of integrating archaeology, heritage, and anthropology in the Maya region.

## 2. The Afterlife of Processualism

Those of us weaned on processual archaeology were trained to theorize process over event, to view "the archaeological record" as a palimpsest of the material fallout from past activities, and most importantly to understand material remains of the past as long-term accretional—rather than episodal—evidence of time's arrow. This sort of logic was deemed to be highly anthropological because, unlike the fetishlike fascination with artifacts and their chronological seriation displayed by our culture historian grandmothers and grandfathers (mostly grandfathers), processualists were more concerned with what artifacts and ecofacts could tell us about living conditions of the past—strategies of hunting, foraging, farming or building cities. Archaeology, we were taught, was about explaining the sweeping changes that had shaped humans and were shaped by them over past millennia.

This process-focused theorizing of the past was remarkably devoid of people and their desires or beliefs [3]. Some processualists adopted a Marxist approach to understanding the past based on the premise that changes in forces and factors of production were easier to monitor archaeologically than were spiritual, ideological, or ontological changes. In retrospect, this premise is factually dubious given the early and overwhelming presence of non-residential, ritual-focused structures in the Maya region and elsewhere [4–6]. To counter processualism, other approaches to understanding the past were proposed, most prominently postprocessualism, which embraced the study of religion and ideology, historical narrative, narrative voice, and the meaning that places of the past have to contemporary people [7]. Likewise, the assertion that gendered understandings of the past were possible [8] sounded an alarm that important constituencies were being systematically omitted from archaeology, both interpretively and at the front end—in the design and execution of research. This reckoning with systemic exclusion foreshadowed the development of Indigenous Archaeology [9].

Contemporary peoples—be they descendants or communities local to archaeological sites—were another neglected constituency that seldom were enfranchised in the archaeological process of research design or interpretation of results. Communities with a strong interest in (and claim to) the material remains left by forbearers simply were not included in the archaeological endeavor beyond the invocation of ethnographic analogy, which constituted a major interpretative strut in places such as Southwestern U.S.A. and the Maya region [10]. Over time, the processual tack morphed into increasingly sophisticated analyses of archaeological materials and environment/subsistence indictors that—while indicative of the impressive maturation of the field in terms of materials science—rendered archaeologists far removed from people of any time period. In the search for deep structures of change, archaeology became an estranged stepchild of anthropology within the Americas.

## 3. Indigenous Critiques of Anthropology

After the 1970s, the discipline of anthropology increasingly faced what might be called a crisis of representation [11] (pp. 50–68). Socio-cultural anthropologists grappled with the erosion of ethnographic authority that occurred as postmodernism broadsided the discipline. Some questioned whether anthropology—the study of humans in all their spectacular diversity through space and time—would survive the challenge. In some cases, the ethnographic study of "my village" was abandoned in favor of multi-sited ethnographies that focused on a process or topic such as migration that could be mapped across space [12]. Another trend within socio-cultural anthropology was to double-down on a study area but within an activist mode—to trade not in knowledge for knowledge's sake but to work towards

social or environmental justice for/with a marginalized community [13,14]. In the end, the ethnographic method—structured conversation—survived the challenges of postmodernism and spread to other topical concentrations, such as cultural studies, global studies, and religious studies, which lack a disciplinary core of methods. From the perspective of Indigenous and settler-colonial studies (more on the latter below), however, even the ethnographic method was suspected as an intervention too far [15–18].

In the Maya region, the mid-century Harvard-supported ethnographic project in Chiapas, México, led by Evon Vogt [19], cast a long shadow. Explicitly engaged with symbolism, ritual, language, and cognition, Vogt and his students pursued a research agenda that couldn't have been farther from the paradigm of processual archaeology. Due to long-durational characteristics of cultural practices in the Maya region—written texts, ritual practices, intentional deposits to dedicate and terminate structures, unambiguous evidence of social inequality, among others—there existed (and continues to exist) a synergistic relationship between Socio-cultural Anthropology and Archaeology. Vogt, in particular, was intrigued by the robusticity over time of ritual practices that he observed in Chiapas with those that could be inferred from archaeological evidence [20]. Tacking between the past and present proved very productive for Maya Archaeologists (and Art Historians) but did it contribute to allegations that archaeologists were freeze-framing Maya peoples to create the "timeless Maya" [21]?

Working in Yucatán in the tradition of Vogt, socio-cultural anthropologist Astor-Aguilera [22] observed the distinctive relationality that exists between objects and people in which objects are perceived to be *communicating devices* rather than passive artifacts waiting to be measured and described. This ontological turn—focusing on how things come into being and the relationships among beings—highlights epistemic contrasts in a sharp manner and brings one to question whether or not archaeological inference in the Maya region stands a chance of being on the mark without significant intellectual investment from descendant communities.

What would a more equitable research arrangement—in which Indigenous and Western scholars worked side by side—look like? In an effort to move beyond polarized and hierarchical contrasts between the Global North and South, Comaroff and Comaroff [23] consider communities of the Global South (sub-Saharan Africa in particular) as participants in other modalities or "alternative modernities" rather than "developing countries"—the latter a rubric that can be equated with infantilism. A critical part of a differently construed relationship between the North and South involves the ceding of scholarly "air time" to intellectuals of and from the South (as well as marginalized populations within the North) in the form of publication outlets and citation patterns. In the process, important critical voices and alternative perspectives surface [24,25].

In the Maya region, Indigenous scholars such as Juan Castillo Cocom and colleagues [26] question the totalitarian fashion in which Maya culture and identity is represented in Western knowledge as situated within the four pillars of Linguistics, History, Anthropology, and Archaeology. Instead of accepting this framing, Castillo Cocom and colleagues [26] (p. 50) invoke *iknal*, a Yucatec Mayan concept that "translates roughly as an extension of social agency, of *perspective*, *presence*, *action*, and *attitude*" (italics in original). In doing so, they practice a form of "ethnographic refusal"—a term introduced by Audra Simpson [16] in response to centuries of mis-representation of Mohawk life by anthropologists. Such critique is part of the ongoing crisis that Anthropology faces—particularly if the discipline continues to conduct research *on* rather than *with* Indigenous peoples. If Maya Archaeology is to be anthropological, then we need to confront Indigenous critiques of Western representation and the corresponding desire for greater self-representation in both the past and the present.

Simpson [16] (p. 97) further notes the critical importance of self-representation, which goes hand-in-hand with political sovereignty. For Simpson, they are two sides of the same coin: "within Indigenous contexts, when the people we speak of speak for themselves, their sovereignty interrupts anthropological portraits of timelessness, procedure, and function that dominate representations of their past and, sometimes, their future" [16] (p. 97). Another critique of Anthropology is based upon its obsessive focus on "the ethnological formalism and fetishism" of difference [16] (p. 97). Although

many anthropologists cite the study of human difference—cultural, biological, and social—as a central pillar of the discipline, it is easy to grasp that the study of *difference* itself might be seen as suspect by Indigenous scholars such as Simpson who views it as a gloss for white superiority [16]. Given the history of racism against Native Americans in both northern and southern hemispheres, this critical framing of difference should not surprise us.

During a time when the call to decolonize methodologies—particularly in reference to social-science and medical research among Indigenous communities—plays an increasingly visible and prominent role [27], what does it mean for Maya Archaeology to be anthropological? How does the study of archaeology, grounded in landscape and place, adopt decolonial methods that enfranchise and benefit local and descendant communities in the study and conservation of the past? Here, I suggest that an archaeology that is attuned to cultural heritage provides a pathway (or ethnoexodus, as Castillo Cocom and colleagues [26] write) towards a future that is more sustainable and can lead to greater accuracy in archaeological interpretations.

## 4. Archaeology and Heritage as Restless Bedfellows

Is there such a thing as archaeological heritage? Does this phrase signify an important subset of tangible heritage? In pondering this oft-used term, I found myself consulting the online Merriam Webster dictionary, where the term "heritage" is defined as "something transmitted by or acquired from a predecessor" or "something possessed as a result of one's natural situation or birth." Certainly, archaeological methods are something that we—as archaeologists—inherit from our intellectual ancestors (although not as a birthright regardless of how many times archaeologists profess to have been born with a trowel in their hands). The material remains that occupy our waking thoughts often are not something that was "transmitted by or acquired from a predecessor" although junior archaeologists do sometimes "inherit" archaeological collections housed in museums and university labs—materials collected by predecessors. One can invoke UNESCO platitudes about the universal value of heritage places that are inscribed on the World Heritage list but one needs to tread carefully through that minefield. Although lofty notions about universal heritage sound unassailable, critiques of the UNESCO concept of universally valued cultural heritage emphasize the overtly Western, high-handed, and monument-centric framework within which this concept has been applied (for more discussion and examples from Çatalhoyuk, Turkey and Western Europe, see [28–30]).

A landscape-based approach to heritage is another option. As inhabitants of a landscape that contains material remains of the past, current residents (regardless of ancestry) do—in a sense—inherit those remains and a responsibility towards them, which might include archaeological research linked with conservation. Would this logic still hold, however, if current residents had established themselves through violent take-over of the land and attempted genocide and removal of original residents? At this point, we move into the realm of settler colonialism—which is a form of exogenous domination that entails displacement of and unequal relations with an original population [31] (p. 1). From the vantage point of those who were unsettled and marginalized by 16th through 19th century population dispersals from Europe, claims of settler colonialists to rights of stewardship over the past can ring hollow [32].

Given these complexities, it is probably advisable to view the two approaches to the past—archaeological on the one hand and heritage-focused on the other—as separate but related approaches. By suggesting this, I am not discounting a connection to old places on a landscape that is not ancestral in any sense, e.g., [33]; but I am stating that such a connection should not be called archaeological heritage. Rather, it is closer to the sensibility of cultural heritage, a subject-focused perception of a connection to something or some practice that is rooted in the past. By keeping cultural heritage distinct from archaeological practice, the two can be held in productive tension—as they assuredly are. The focused positionality of heritage can provide a voice and a platform for those who otherwise may be marginalized from archaeological research. Re-centering archaeological practice

in respect to heritage issues opens the discipline to community perspectives on heritage and local priorities for heritage conservation.

In the 16th through 18th centuries, emigrants left a place known as Castile to invade and colonize the Americas (particularly the southern part). As is well-known, they emanated from (and in some cases sought to escape) a place in which there was zero tolerance for religious diversity. Treaty negotiations with original inhabitants of the Americas were not on their minds, contrary to the case with later British colonists to the north. The roughly 300-year colonial period within a region that 20th-century anthropologists came to call Mesoamerica was marked by successive and violent efforts to dispossess Indigenous peoples from their land, strip away any and all political sovereignty, de-legitimize cultural practices (particularly religious beliefs), and erase pre-Columbian history where ever possible. As Veracini [31] (p. 3) points out, the goal of settler colonialism is its erasure—arriving at the point in time at which settlers assume "native" status. Writing from an Indigenous perspective, Sherman Alexie [34] (p. 95) declares that "In the Great American Indian novel, when it is finally written, all of the white people will be Indians and all of the Indians will be ghosts" (cited in Tuck [2] (p. 647)).

A variant on the erasure of settler colonialism is the Latin American myth of *mestizaje* or the notion that European and Indigenous-derived characteristics—everything from genotypes to philosophies of statecrafts—are so thoroughly inter-mixed as to be inseparable [35]. While it is true that the two are inseparable parts of a whole, many Indigenous communities remain distinctively separate spatially, culturally, and tragically economically. Within México, Bonfil Batalla [36] exploded the myth of a cultural heritage that is composed of centuries-old mixing of cultures with the publication of *México Profundo*.

Within Mesoamerica, settler colonialism promoted an intimate, social hierarchy while also engineering dispossession from land and from landscape features that revealed a deep precolonial imprint. This kind of heritage distancing [37] was coded into the educational curriculum beginning with primary schools and continued to ramify relentlessly through adolescence and adulthood. This estrangement from deep heritage as a strategy of settler colonialism is made more obvious by comparison to other places not subjected to settler colonialism—such as China—in which a connection to deep heritage is widely shared, albeit expressed with a range of feelings from deep emotion to casual comment.

The point of this section on the restless intimacy between archaeology and heritage is to urge a critical evaluation of archaeology in relation to cultural heritage and to take the long-term, knock-on effects of settler colonialism seriously. Historically, if not a handmaiden of colonialism, archaeology has been a beneficiary of policies abetted by regimes of settler colonialists, particularly within Mesoamerica. This beneficial relationship expanded as heritage tourism grew through the twentieth century to become a significant part of national economies [35,38]. Corollary to this growth is increasing recognition on the part of archaeologists that, for the most part, Indigenous peoples of Mesoamerica have been estranged from cultural heritage that is linked to landscape. Such estrangement is indicated through limited access to archaeological sites or rights to perform ceremonies within the limits of sacred places that currently are controlled by national or state agencies [39]. Further estrangement happens through commodification of heritage tourism in a manner that provides scant benefit to descendant communities and the destruction of sacred places despite the protests of local communities.

Knitting together the terms archaeology and heritage will not remedy this situation but only prolong the restless nights of these ill-suited bedfellows. A more productive approach is to recognize that archaeology and heritage are two very separate ways of relating to the past and to work proactively to ensure that descendant/local communities have a right to exercise authority over decisions regarding their heritage. When archaeologists work to safeguard this process, we engage in methods otherwise known as decolonization [27]. For all of these reasons, my answer to the question posed at the beginning of this section—"Is there such a thing as archaeological heritage?"—is no. They are two very distinct ways of relating to the past and both have been complicated immensely by factors of settler colonialism.

## 5. Return to Process within Archaeology

Emphasizing process within archaeological methods (leaving theory aside for the moment) slows things down. Fieldwork becomes a process rather than an event, which allows time for consultation, collaboration, and other kinds of community participation [40,41]. As colleagues in Australia and Oceania wrote decades ago, process-based practice acknowledges that building trust with local and descendant communities is an important part of practicing archaeology ethically and sustainably [42–44]. At the end of a field season, rather than asking each other "What did you find?" or fending off the common query from interested laypersons, "What is the coolest thing you have ever found?", attention might be focused on anthropological questions that probe how an archaeological project is embedded within a heritage landscape or the specifics of how an archaeologist works with descendant/local community members to whom we ultimately are accountable. These concerns are central to our discipline but historically have existed as a shadowy backdrop concealed by the zeal of archaeological discovery.

Reflecting on the process of fieldwork [45] need not detract from archaeological discovery. Rather, the process by which we get to discovery and subsequent co-production of knowledge follows a pathway that is more richly informed due to input from multiple sources. As knowledge about past and current landscapes becomes more routinely co-produced, archaeologists will need to step back from territorial claims on ideas, artifacts, and sites; this may be the most challenging part—ceding some control [more discussion of this in [46].

Within the Maya region, a process-focused archaeology would be more anthropological in three ways: it would reckon with Indigenous critiques of anthropology discussed above, recognize Indigenous authority over research and interpretation, and work with communities to investigate, interpret, and conserve remains of the past (examples provided in section to follow). This kind of practice is not a "move to innocence"—a term that Tuck and Wang [47] (p. 9–28) use to refer to decolonizing efforts that are largely metaphorical and achieve no real or positive change (or in their opinion, do not result in "repatriation of Indigenous land and life" [47] (p. 21). Process-focused methods will change archaeology and greatly benefit local communities. Community benefit, in fact, is a good yardstick by which to measure whether "working with communities" is only the self-congratulatory, avant-garde turn critiqued by LaSalle [48] or represents real change from business as usual.

For several reasons, Maya archaeology is far from embracing what Tuck [2] (p. 644) refers to as a "methodology of repatriation." There are strong headwinds; institutional and structural changes are needed and will take time. Funding agencies—especially the National Science Foundation—need to reckon with the importance of time-consuming processes that render archaeological research more ethical and responsive to community. Tenure-review committees at U.S. colleges and universities need to acknowledge the value of a longer cycle of researcher investment in community and reward such investment with tenure and promotion. Finally, co-management arrangements in which government-permitting agencies share authority over places of heritage with local communities—particularly those within Latin America—need to become the default instead of over-centralized control of the past. The proliferation of community museums within México, and particularly in Oaxaca, indicates that a change in which local communities have increased authority over the representation of their past is reachable [49].

Any effort to bolster Indigenous authority over self-representation is a step in the right direction. Here, the connection with anthropology is woven into the fieldwork process as well as interpretive design. In large and well-funded projects, socio-cultural anthropologists may work side-by-side with archaeologists in cultivating community relationships but the two should assume equal importance. Subjectivities that are expressed through the idiom of cultural heritage or other knowledge systems become another interpretive strand to be braided—as Sonya Atalay [40] (p. 76) has written—into narratives of the past. This pathway is not without conflict and admittedly is more time consuming and uncertain but it is not only desirable on the basis of ethics and social justice, it will lead to interpretive narratives that are better informed. As historian John Hope Franklin [50] noted in reference to the

inclusion of African-American voices in U.S. history, a narrative created on the basis of more than one perspective is a more accurate history.

## 6. Integrating Archaeology, Heritage, and Anthropology in the Maya Region

Within the last decade or so, the National Science Foundation has nodded towards the need for scientific research to include "Broader Impacts" to society. Archaeological proposals, ostensibly all about the past, are required to show relevance to contemporary issues or challenges. Acceptable relevance might include enfranchising marginalized populations into the research process or disseminating knowledge about the results of archaeological research to communities proximate to a research site. As a reviewer of NSF proposals, I can vouch for the fact that it is rare to see a Broader Impacts statement that is inspired or particularly creative—most of the intellectual "juice" seems to be expended on traditional research design and methods. Why do archaeologists not take "Broader Impacts" to society seriously?

The answer is multi-dimensional. First, lack of competence and creativity in designing plans of broader impact likely is indicative of the nature of training in anthropological archaeology that is offered within the U.S. The rift between archaeology and socio-cultural anthropology in many academic departments throughout the U.S. has left archaeologists ill-prepared to work with people. Second, there is a misconception that community archaeology does not lead to journal publications but assuredly this issue of *Heritage* goes a long way towards dispelling that idea. Finally, archaeologists are uncertain about "getting credit" for time spent cultivating a relationship of trust with a community. While this might be a legitimate concern for graduate students who are "under the gun" to complete their dissertation fieldwork, professionals—at any stage of their career—should expect to make an investment in a place in order to generate a working relationship. Socio-cultural anthropologists engage in decades-long programs of ethnographic fieldwork for just this reason. By abandoning the helicopter approach to fieldwork and taking broader impacts seriously, archaeologists have the opportunity to gain deeper perspective on local landscapes and their inhabitation.

Whether this involves taking Maya archaeology in a more anthropological direction or attending to issues of heritage at field sites, such initiatives—when seated within more process-focused field methods—intensify interaction with local communities (see Hutson et al. in this issue). As discussed elsewhere [11], the solely dyadic relationship between archaeologists and things/places of the past is dissolved in favor of a triadic structure that includes peoples/communities/constituencies/heritage stakeholders (whichever term you prefer) as the third member of the triad. As Charles Hale [13] has written in reference to activist socio-cultural anthropology, this is a complicated and potentially compromising place to occupy. Mistakes will be made and opportunities will be missed but the potential for creating long-term research partnerships is considerable, which makes the investment by archaeologists extremely worthwhile.

An activist socio-cultural anthropologist or cultural geographer may interact with communities about a burning issue such as a land-claim settlement and then move on after the land claim is settled [14,51]. But archaeological sites are fixed on a landscape—they do not move on. They either persist in place or suffer deterioration due to natural causes or purposeful destruction. Because of this fixity, I suggest that the following two matters are of great and lasting importance: (1) accepting the triadic structure of our profession (which includes communities, archaeologists, and remains of the past) and (2) establishing long-durational relationships with communities close to places of archaeological research.

A. **Programs of** *The Maya Area Cultural Heritage Initiative* **and** *InHerit: Indigenous Heritage Passed to Present*

Through a combination of grant-writing, donations from private foundations, and support initially from Boston University and then from the University of North Carolina-Chapel Hill, I have had the opportunity to explore variations on this triadic relationship (see www.in-herit.org for details). Explicitly anthropological and heritage-focused, programs based on this triad have encompassed a

range of educational, entertaining, and experiential activities transmitted through radio shows, school workshops, and archaeological excavations. Just about any medium of transmission available in the Maya region has been utilized at some point in time. Some of the most successful programs have been radically out-of-the-box and related to archaeology, sensu stricto, in only the most tangential way (at least that is what I thought at the time). At point of contact, these heritage programs yielded benefit to involved communities, but measuring the long-term impact of these programs is another matter altogether. Frankly, I do not know whether we increased university admittance among young participants or how many archaeological structures—destined for the bulldozer or targeted for looting—were saved. But I do know that the heritage programs were humbling and learning experiences for me and for my staff. These engagements with reality forced us to push against the edge of what archaeologists generally know about communities and their social landscapes.

For instance, community mapping in the Guatemalan Highlands resulted in recording shrines (past and present) along with oral histories of shrine locations. In one community, the information was accepted and placed under seal by the town council due to its perceived sensitivity [46]. Ceding control over such data is antithetical to the goals of archaeology yet (and the irony does not escape me) it was the ethical course of action and one that respected the sovereignty of Indigenous communities.

Another heritage program involved the creation and radio performance (in both Q'eqchi' and Spanish languages) of heritage-focused skits. Coordinated by a Petén-based nonprofit called ProPetén, the idea was to project an ideal world in which K-12 school teachers has resources to teach about the fabulous archaeological sites of the Petén and take students on field trips to Tikal and other sites groomed for tourism (for more details, see [11], p. 115–121). After the *radionovelas* aired, ProPetén convened focus groups in small towns where community members had listened to the radio shows. The transcripts of those focus groups and accompanying questionnaires are very sobering and reveal a large group of overlooked young females who—with extremely limited formal education—were very curious about the old places on their landscape and felt strongly that they should be conserved for future generations. Their voices are marginalized from national and even local discourse. Heritage programs may amplify seldom-heard voices, but converting that amplification into meaningful change in the lives of young rural women is far more challenging, which highlights a limitation of such initiatives.

With the success of the *radionovelas* in the Petén, we decided to expand the idea to the northern lowlands (with changes in the content and language of the radio shows). Since our resources were dwindling, we had to decide whether the script was to be written and performed in Spanish or Yucatec Mayan. Because of our commitment to the survival of Indigenous languages, we chose to broadcast in Yucatec Mayan but, by doing so, excluded a very large Spanish-speaking constituency who either identify as Yucatec Maya but, as children, were not taught the language or do not identify as Yucatec Maya but live in and around Felipe Carillo Puerto (Quintana Roo, where the radio shows were broadcast) and are intensely interested in conserving old places (more details of this program in [11], p. 177–179).

Language, culture, and literacy are entangled in complicated ways that can be under-appreciated. One of our first efforts to boost Indigenous languages was based in the Toledo District of southern Belize. We compiled a small booklet called "Seeing our Ancestors" that was translated into Mopan and Q'eqchi' Mayan. An academician's idea of a "user-friendly" booklet, we generated far too much script with far too few images (a graphic novel would have been far more impactful). The local community—completely conversant in Mopan or Q'eqchi' or both—struggled to read the text in a language they rarely saw in written form.

The heritage programs sponsored by MACHI and InHerit were always grass-roots and tailored to place but nonetheless, there was a tendency to homogenize. After the gifted artist Carin Steen produced a coloring book for young Ch'orti' children with a few sentences of Ch'orti' text on each page, I imagined that we could use the graphics in other parts of the Maya region and simply swap out the linguistic part. Wrong. Images, dress, archaeological sites, and local ritual activities were not generalizable and did not resonate outside of the Ch'orti' homeland. With this realization, I began

to grapple with the cultural distinctiveness of locality within the Maya region, a characteristic that is not surprising to socio-cultural anthropologists. But for an archaeologist—trained to think about The Ancient Maya as a monolithic thing—the gap between contemporary reality and archaeological imaginaries opened into a yawning chasm.

Over time, my research focus shifted to northern Yucatán. I became intrigued by the karst landscape that had been successfully peopled, farmed, and governed until ruptures caused by the Spanish wars of the sixteenth century [52]. The centrality of sinkhole features (*cenotes* and *rejolladas*) to settlement and farming—particularly in the past—is inescapable. The porosity of this karstic terrain also highlights the vulnerability of the underlying aquifer to pollution. More recently and with funding from the National Geographic Society, we have been able to work with middle-school teachers in nine communities around Valladolid, Yucatán, to create a *cenote*-focused curriculum for teachers and interactive experiential learning for students [53]. A workbook—the culmination of the project—highlights the importance of *cenotes* as sources of clean, fresh water that support a complex ecosystem as well as the urgency of their conservation [54]. The workbooks also highlight the visibility of *cenotes* in two of the four known pre-Columbian codices—the Codex Madrid and Dresden.

The middle-school and college-age students who participated in the classroom workshops resulting in the workbook displayed an impressive awareness of the beauty and fragility of *cenote* landforms and of the dangers posed by pollution. On the other hand, few students were aware of the codices produced by their ancestors and stored, for the most part, in European libraries and archives. Although recognized globally as irreplaceable treasures of world heritage, Maya codices do not make their way into Yucatec school curricula. The past five hundred years of settler colonialism has estranged Indigenous peoples not only from their landscapes but also from their intellectual history of book production. There is little space for discussing Indigenous heritage within an educational system that is predicated upon racism and maintaining certain forms of colonial domination. For the most part, history is taught as beginning with the 16th-century arrival of Spaniards and missionaries are portrayed as having worked tirelessly to eradicate the work of Satan (which included painted manuscripts). Time and again, I saw students marvel at the codex facsimiles upon their first exposure to these ancient books. Gabrielle Vail ran workshops on the codices, pointed out the many representations of *cenotes* in the Madrid and Dresden codices, and taught students to identify deities painted on the pages of books produced by their ancestors. Throughout these workshops, the injustice of this estrangement was inescapable. One can be excused for hoping that it is only a matter of time before these students become adults and petition through diplomatic channels to have their books returned from libraries and archives in Dresden and Madrid.

*Cenotes* are central to cultural heritage in Yucatán. Perhaps my co-director, Iván Batún Alpuche put it best when he described the goal of this bio-cultural heritage program as the repatrimonialization of *cenotes* (returning authority over cenotes to communities as part of their legitimate patrimony or heritage). From his perspective, this program should work towards *cenote* sovereignty or the authority of local communities to manage, conserve, and protect their water supply and associated bio-cultural ecosystems. Of course, no *cenote* is an island unto itself—all are connected to the underground aquifer. This knowledge is deeply seated within Yucatec Maya ontologies and traditional ritual practice and also a central tenet of karst hydrology. As such, it provides a great example of the convergence of different knowledge systems. Thus, the challenge expands; to be effective, a program of heritage conservation must include all *cenotes*. Such a large goal is overwhelming but an important point of this example is that there are heritage-linked issues that are bigger than archaeology. We need to embrace this expansiveness rather than shy away from it.

## 7. Conclusions

By imagining an anthropological archaeology that is attuned to Indigenous issues of cultural heritage, Maya archaeology shifts into a hybridized practice that blends anthropological emphasis on contemporary people with their perception of things, places, and landscapes of the past. This imagined

archaeology takes account and is respectful of the myriad ways in which the subjectivities of cultural heritage are locally seated and it places local ideas in productive tension with archaeological ideas and anthropological concepts. Such a critical lane shift represents a change from product-oriented goals to process-focused collaborative research. This shift allows archaeology to shed epithets such as neocolonial and extractive while embracing more inclusive and multi-braided approaches to knowledge production. Such a transition also requires attention to a balanced calculus of benefits—that is, attending to who is benefitting from archaeological research. My late colleague Dorothy Holland—a champion of participatory research—often stated that one can gauge how truly participatory a project is by who is seated at the table when decisions are made, deals brokered, and budgets allocated.

This shift also recognizes what I have called elsewhere [11] (p. 5) the triad of agents: archaeologists, local/descendant/concerned communities, and the material remains of the past (aka non-human agents). Instead of the intense dyadic relationship between archaeologists and materials of the past, community-collaborative approaches dimensionalize that space into three dimensions. This shift in geometric form opens a world of opportunities for archaeology in the realms of research design, execution, interpretation and importantly heritage conservation. While there are challenges and uncertainties associated with this evolving epistemology and practice, there also is transformative potential. Here, I have attempted to trace how we got to this place and why the path forward should look very different from our grandfathers' Maya archaeology.

**Author Contributions:** This article was written solely by the single author who accepts all responsibility for its content and limitations. All authors have read and agreed to the published version of the manuscript.

**Funding:** Over the two decades during which the heritage programs discussed in this article were implemented, funding was provided by private donors, family foundations, the Archaeological Institute of America, the National Geographic Society, and the U.S. State Department.

**Acknowledgments:** Over the many years that MACHI and InHerit have operated, many talented people have lent their genius to the creation and implementation of heritage programs. I mention a few names here but please consult acknowledgments in McAnany [11] for a more comprehensive accounting, especially of earlier programs. None of these programs would have gotten off the ground without my Program Directors. From present to past they include: Dylan Clark, Gabrielle Vail, Claire Novotny, Sarah Rowe, and Shoshaunna Parks. Collaborating with our partners in the Maya region, together we wrote grant proposals, reports, journal articles, newsletters, and much more. For our most recent endeavor in the Maya region—the Yucatec cenotes project—we are grateful for the support of the National Geographic Society (Grant No. HJ-147E-17) and to our Yucatec Program co-Director, Iván Batún Alpuche for his vision of Yucatec Maya heritage. Dylan Clark and Khristin Landry Montes expertly guided field implementation and worked closely with middle-school teachers who bravely piloted the curriculum and offered immensely valuable feedback. The Research Labs of Archaeology at UNC-Chapel Hill have steadfastly supported the InHerit programs and provided space and administrative support for their continuation. I thank peer reviewers of this article for their insightful comments and suggestions.

**Conflicts of Interest:** The author declares no conflict of interest.

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
