# Peer review of "Imagining a Maya Archaeology That Is Anthropological and Attuned to Indigenous Cultural Heritage"

_heritage, doi:10.3390/heritage3020019_

Round 1
Reviewer 1 Report
Review of “Imagining a Maya Archaeology that is Anthropological and Attuned to Indigenous Cultural Heritage” by Patricia A. McAnany
This is a very important call to action by a senior scholar in the field of Maya archaeology. McAnany has committed decades of effort to recreating how archaeological practice can be conducted in collaboration with contemporary Maya speakers, and how the results can be made relevant to their lives. In this piece she explains why Mayanists remain stubbornly traditional in terms of how they practice archaeology and the types of community engagement they largely choose not to avoid despite trends in the opposite direction within North America and elsewhere in the New World. She also briefly summarizes some of the main critiques of archaeology by indigenous scholars, which emphasize the need for sovereignty over the past as well as the present. Her many years of experience drawing archaeological sites, materials, and knowledge into contemporary heritage conversations provide solid footing for a theoretical exploration of how it is impossible to neatly fit ‘archaeology’ into current ‘heritage’, but rather the complexities of their interpenetrating but distinct histories and agendas must be acknowledged. Finally she concludes with a call for a methodological shift from understanding fieldwork as an event to viewing fieldwork as a process—one which takes more time than in the past, but which opens up space for trust-building and collaboration in the meantime. This section is enhanced with specific examples from her many years directing MACHI and InHerit, which should be highly instructive to Mayanists.
This is a strong piece of writing and my suggestions are mostly limited to requests for the author to elaborate a point, or define a concept. Please see the numbered suggestions below.
I have two more substantive suggestions. The first is to address why archaeologists must do this work of community engagement rather than hiring or adding a cultural anthropologist to their projects and thus going about their usual way of doing fieldwork. I believe the author agrees that archaeologists need to do this work, but I also know many readers of Heritage will think the most “appropriate” course of action would be to employ a trained cultural anthropologist. Of course there is no one right way to do these things, but I expect readers will want to know the author’s opinion.
The other substantive edit would be to the second section of the paper, Archaeology and Heritage as Restless Bedfellows. It is both brief and complex. The author is deeply enmeshed in these issues and has been for some time, so it is interesting to read her thoughts on how the two fields don’t actually naturally converge. But to me, this section could be tightened up while still retaining the messy complexity that is ultimately the point she is trying to make. Perhaps by returning to the question she poses at the start of the section, “Is there such a thing as archaeological heritage?” and attempting an answer, some of the various threads made visible in this section can be brought together a bit more for the reader who is less well versed in these complexities.
Specific comments:
70—The inclusion of the critique that gendered voices had been left out of archaeology is an important reminder that feminist archaeology helped generate the groundwork for indigenous arch
81—The observation that processualism morphs into materials science, and thus becomes even more removed from descendent communities and anthropology as a whole is very insightful, I love this
136—“the indigenous critique of Western representations” Is there a single coherent indigenous critique? Maybe allow that there are multiple? As multiple Maya communities, perspectives, etc…this is a place where a bit more elaboration would be helpful
147—Are there citations for indigenous critique of the study of difference?
161—Please clarify the language about the connection between inherited materials and UNESCO—many archaeologists do inherit collections of material remains they become deeply invested in, so are you speaking about not inheriting methods or materials or both? Also, perhaps the readers of Heritage are familiar with critiques of UNESCO, but I think it would be worthwhile to either add a couple of sentences here to elaborate or more citations to the archaeological communities' critiques of UNESCO
171—In regards to “current residents”—It would be helpful to provide more context on how many Maya speaking people live near archaeological ruins
175—Define settler colonialism
180—“By suggesting this, I am not stating that feeling a connection to old places on a landscape that is not ancestral in any sense is not genuine or authentic” too many nots
189—I suggest you switch the order of the paragraph; state goal of settler colonialism then explain its roots in past
225—Could you expand what estrangement means in the Maya area? Its not just being unable to enter sites…It includes 500 years of the destruction of sites with remnant vestiges still visible, the commercialization of Maya archaeological heritage in ways that exclude most small communities, etc etc
233—This is perhaps the most important contribution of the paper in this reviewers perspective and it deserves a bit more elaboration, perhaps with specific examples of how a Mayanist can shift their understanding of fieldwork from an event to a process
282—In the discussion of why broader impacts in NSF proposals are not more creative—Could you flesh this out, and dig deeper? Ultimately taking these requirements seriously would change the field in ways you argue are necessary. Could factors such as a perception that such outcomes are not part of tenure reviews, or don’t lead to publications, be important? Use your platform as a senior scholar in our field to argue for a reconceptualization of this component of NSF funding
287—Please clarify who the players are in the “triadic structure”? Are they the same as the “triad of agents” mentioned on line 408? If so, please make explicit.
300 InHerit case study should get its own section heading?
313—Heritage programs were humbling experiences for whom? For her? For community?
382—Hope for the future repatriation of codices is laudable but is that really the reason young Yucatec speakers don’t know about the codices? I think the author could dig a bit deeper here and state something about the institutionalized power at play in modern Latin American education systems, which coupled with racism, and poverty, are at the heart of why many modern Maya speakers know so little about their ancestral heritage
385-Please define repatrimonialization
Author Response
Below, I list how and where I address the comments and suggestions of Reviewer 1.
L. 290-292, I discuss how archaeologists might work side-by-side with socio-cultural anthropologist or go it alone but that either way, the two endeavors should be of equal importance.
L.244-246, I summarize the section on Archaeology and Heritage as Restless Bedfellows by discussing how settler colonialism has complicated the relationship between the two.
L. 75-80, I call out Indigenous Archaeology as engendered by discussions of gender in archaeology (as pointed out by the reviewer).
L. 87-100, I have revised the text and pluralized Indigenous Critiques of Anthropology
L. 140-150, I clarified that Simpson is the author of the Indigenous critique of the study of difference
L. 160-175, I clarified the wording of what is and what is not heritage and how heritage is different from archaeology. Towards the end of the paragraph, I cite specific studies/critiques of the UNESCO framework and ethos.
L. 176-177, here the reviewer asked for specific numbers of residents living near archaeological sites in the Maya region but I think that he/she misconstrued the meaning of this sentence. This sentence is part of a general discussion about how people relate to the past in other-than-archaeological ways and was not meant for refer specifically to the Maya region.
L. 181-183, I define settler colonialism as requested
L. 187-189, I removed several negative "not"s from this sentence--good catch.
L. 204-205, again on settler colonialism, which is now defined above so, I have not changed the placement of this sentence.
L. 232-236, I expanded the many ways in which local people are estranged from archaeological sites
L. 268, in the section on process-focused archaeological methods, I communicate that examples of this "are provided in the section to follow"
L. 310-322, I have added a detailed discussion of why archaeologists might not take NSF Broader Impacts to Society statements seriously--good suggestion
L. 338, triadic structure defined in parentheses
L. 342, subheading for heritage programs created, as suggested
L. 357, clarified who was humbled by the heritage programs
L. 421-425, provided more discussion of why Maya codices are not taught in the middle schools of Yucatán
L. 435-436, defined repatrimonialization
Reviewer 2 Report
Summary
The author states that the goal of her article is to “imagine a different kind of Maya archaeology,” in which the focus is not on producing a product but is, instead, on process. Specifically, the author envisions an approach to Maya archaeology that is both anthropological and attuned to heritage perspectives. The article begins with a concise history of the paradigmatical war in archaeology between processualism and postprocessualism, noting that in a quest to become increasingly scientific, processualism moved farther and farther away from the people it claimed to study and estranged archaeology from anthropology, particularly in the US. The author then reviews anthropology’s internal battle for its soul brought to the field by the forces of postmodernism, noting that the ethnographic method ultimately prevailed. At the same time, anthropology faced critiques from Indigenous scholars who disparage anthropologists for conducting “research on rather than with Indigenous peoples,” raising a call to decolonize anthropological methods. The author then turns to the notion of archaeological heritage and confronts the issue of settler colonialism, suggesting that it is “advisable to view the two approaches to the past—archaeological on the one hand and heritage-focused on the other—as separate but related approaches.” The final sections of the article call for a “processed-focused [Maya] archaeology” that is more anthropological: “it would reckon with the Indigenous critiques of anthropology…, recognize Indigenous authority over research and interpretation, and work with communities to investigate, interpret, and conserve remains of the past.” The author highlights examples from her own research, approaching these issues in Guatemala, Belize, and Mexico over the course of the past several decades. Ultimately, she proposes that by “imagining an anthropological archaeology that is attuned to Indigenous issues of cultural heritage, Maya archaeology shifts into a hybridized practice that blends anthropological emphasis on contemporary people with their perception of things, places, and landscapes of the past.” This incorporation of Indigenous perspectives is the one third of a “triad of agents” rounded out by archaeologists and the material remains of the past.
Review
There is much to like about this article, and it is worthy of publication. It is well written (although I note a few typos below), well organized, and thought provoking. It manages to be reflexive and forward looking, without becoming (too) preachy. I think that Section 4, “Archaeology and Heritage as Restless Bedfellows,” is perhaps the strongest section, particularly its confrontation with settler colonialism in Mesoamerica. I appreciate the fact that the author describes her approach as “aspirational,” at least in the abstract to the paper, because many of the changes she is calling for will require institutional and structural changes that will take time. This is perhaps an unavoidable weakness of the paper, but one that the author could address more explicitly. Much of the envisioned Maya archaeology outlined here would struggle in the face of practical considerations such as the expectations of funding agency, which likely do want to see a “product,” governmental regulations and procedures that may not tolerate a slow “process” that “allows time for consultation, collaboration, and other kinds of community participation,” as well as tenure and promotion clocks and committees that might prioritize publications over multi-season negotiations toward an acceptable research design. The author touches upon another issue in her discussion of the NSF’s incorporation of “Broader Impacts” that is an important consideration here. I, too, have reviewed many NSF proposals and the vast majority tack on some short and shallow discussion of a community engagement initiative as the broader impact. While the author suggests this a product “of the limited nature of professional training in anthropological archaeology that is offered within the U.S.,” I would rephrase that to say simply “the nature of training.” Few graduate programs until recently trained their students in issues of community or public archaeology because that wasn’t a consideration at the time.
Specific Comments
Note that some new paragraphs have a double-tab indent, while most have a single indent.
Line 58: I’m confused by the phrase “our culture historian great grandmothers and fathers (mostly fathers)...” and the author’s return to this metaphor in the closing line of the article where she say says, “our grandfathers’ Maya archaeology” (Line 416). Does the author mean “great grandmothers and great grandfathers” in the first case? I would think the easiest change would be to say “our culture historian grandmothers and grandfathers (mostly grandfathers)…” in Line 58; that makes the last line consistent with the analogy.
Line 63: To say that “Processualism adopted a Marxist approach” glosses over the diversity of voices within the paradigm. Perhaps say “Some processualists adopted…”
Line 197: “it’s erasure” should be “its erasure”
Author Response
Comments of Reviewer 2 were very helpful and I have responded to all her/his suggestions.
1). Regarding the criticism of the realism of this aspirational thinking and the time it will take to see real change for several institutional and structural reasons....In response, I address these issues point by point in lines 263-274 (new additions to text).
2). In response to recommendation to revise text on lines 293-295 regarding training of U.S. graduate students, I have embraced wording suggested by the reviewer.
Specific comments:
3). I have removed (hopefully) all double indentations at the beginning of new paragraphs.
4). Line 58, I have revised text as per reviewer's suggested wording.
5). Line 63, I qualified the statement to read "Some processualists..." as per reviewer's suggestion.
6). Line 197, I removed the apostrophe in "it's".
Reviewer 3 Report
This is a provocative contribution that takes the valid position that archaeologists should be more attuned to local voices. Developing a short critique of Archaeology and Anthropology in general that carries a personal view that may be difficult to fully substantiate, the author opines that there is space to conduct research with local and Indigenous peoples. Considering experience in the Maya area, data are marshalled to show how local peoples have been estranged from their own cultural heritage. Positive results can come from engagement and collaboration with descendant and local communities whose knowledge of the features of the archaeological landscape is a direct link to the heritage of the place. In this call for the co-production of knowledge, common ground is envisioned where the process of anthropological and archaeological fieldwork promotes an integrated design that promises to create an innovative long-term partnership for interpretation, one that is argued will be multi-faceted and respectful. To this we can all be encouraged.
Author Response
This new review arrived after I began revisions.
The reviewers does not request any changes/additions but does note that my reflections/insights on the field of archaeology may be difficult to check with sources. While it is the case that I put things together in a new way, I have made every effort to back up statements with references cited wherever possible.
Reviewer 4 Report
I really enjoyed reading this paper. It is very well written.
My only slightly critical comment is about context and recognition of several decades of work globally to wrestle with substantively similar issues. The author implies that the critique of processual and post processual archaeology and the suggestion of a 'process focused practice' is new. The issues discussed are relevant not only to Mayan archaeology but rather archaeology as whole and especially in colonized countries and has been a point of discussion for over 2 decades amongst archaeologists and Indigenous land owners/scholars (e.g. Marshall, Y 2002 Community Archaeology. Special edn World Archaeology Vol 34:2; Langford R.F 1993 Our Heritage Your Playground, Australian Archaeology 16 pp 1-6; Torrence R and Clarke A 2000 The archaeology of difference: negotiating cross cultural engagement in Oceania, Routledge).
The author could have acknowledged this in the introductory section and section 5 without detracting from the value of reflecting on this and the importance of producing locally tailored approaches to the Mayan context.
Notwithstanding the comment above, the paper is a well articulated contribution from the region and I think it is worth publishing.
Author Response
Great suggestion to include references to community-based archaeology in Oceania and Australia. Marshall's work is well known and I have referenced her Community Arch article as well as Robin Torrence's 2000 publication and Langford's 1983 prescient piece. I nod to them in the introduction and return to cite specifically in section 5 (as suggested).